# Conventionalism, Cosmology and Teleparallel Gravity

Laur Järv  and Piret Kuusk *

Laboratory of Theoretical Physics, Institute of Physics, University of Tartu, W. Ostwaldi 1, 50411 Tartu, Estonia; laur.jarv@ut.ee

* Correspondence: piret.kuusk@ut.ee

**Abstract:** We consider homogeneous and isotropic cosmological models in the framework of three geometrical theories of gravitation. In Einstein's general relativity, they are given in terms of the curvature of the Levi-Civita connection in torsion-free metric spacetimes; in the teleparallel equivalent of general relativity, they are given in terms of the torsion of flat metric spacetimes; and in the symmetric teleparallel equivalent of general relativity, they are given in terms of the nonmetricity of flat torsion-free spacetimes. We argue that although these three formulations seem to be different, the corresponding cosmological models are in fact equivalent and their choice is conventional.

**Keywords:** philosophy of spacetime; Friedmann cosmology; teleparallel gravity; symmetric teleparallel gravity

## 1. Introduction

At present, the ΛCDM model is considered as the most adequate large-scale description of the visible universe. It is based on the Friedmann solution of equations of Einstein's general relativity (GR) with cosmological constant Λ, which is considered to describe hypothetical dark energy, and also contains hypothetical dark matter. The essential parameters of the model can be estimated from observations with ever-improving precision.

However, not everybody acknowledges the success of the ΛCDM model as there are several observational results that are difficult to accommodate in it [1]. The hypothetical dark energy was introduced to explain a totally unexpected discovery of accelerating expansion of the universe during the last 6 billion years. The equally hypothetical dark matter was introduced to explain the observed rotation velocities of stars in galaxies, and is also required to give an account of structure formation and gravitational lensing. However, particles of dark matter have up to now never been observed and they cannot be identified with any particles from the standard model of particle physics.

There are several directions to understand the situation. Physicists who are used to investigating different theories of gravitation are inclined to modify the Einstein general relativity for obtaining predictions close to observations without using hypothetical entities. Philosophers of physics are eager to spot the roots of difficulties in the standard model of cosmology. We are not going to give an overview of the modified theories of gravitation since there is already a wealth of literature on that topic [2,3]. Instead, we will discuss some features of current cosmological models that raise questions. We shall proceed from a recent paper [4], which claims that the standard model of cosmology is in a great part conventional. Conventionalism in physics tries to separate those parts of theories that do not describe real properties of objects under consideration but are simply definitions or conventions that can be replaced by different ones, as far as observational results are retained. The idea was implicitly presented by Duhem [5], elaborated by Poincaré [6] and explained in detail by Popper [7].

The present paper considers the contemporary standard model of cosmology from the point of view of conventionalism, putting focus upon the alternative geometric formulations. We consider theories where gravitation is given not in terms of curvature of



the connection as in general relativity (GR), but in terms of torsion or nonmetricity of the connection. If the curvature and nonmetricity of the corresponding connection are taken to vanish, but the torsion is not, it is known as the teleparallel equivalent of general relativity (TEGR) [8,9]. If the curvature and torsion of the connection are taken to vanish, we obtain the Symmetric teleparallel equivalent of general relativity (STEGR) [10]. The Lagrangians of these three theories differ only by a total derivative term, so their local equations of motion coincide, and one can envision a geometric trinity of gravity [11,12]. A reader interested in the mathematical details of teleparallel theories can refer to reviews like [13,14]. Let us mention that an analogous set of equivalences can be also shown in the nonrelativistic case [15,16], as well as in extended $f(R)$-type case [17], or including both torsion and nonmetricity simultaneously [18].

The paper is structured as follows. In Section 2, we briefly review the paper by Merritt [4], where the conventions of $\Lambda$CDM cosmology are compared with those characterized by Popper [7]. Then, we the discuss problems of theoretical and interpretational equivalence in physics as presented by Weatherall [19], Dürr [20] and Coffey [21]. In Section 3, we flesh out how the spatially flat Friedmann $\Lambda$CDM model is described in GR, TEGR and STEGR. At the end, Section 4 is devoted to a discussion that summarizes the results and outlines some avenues beyond $\Lambda$CDM which deserve a closer look.

## 2. Conventionalism in Physics and Cosmology

### 2.1. Merritt on Conventions in the $\Lambda$CDM Model

Merritt [4] proceeds from Popper's idea that science as distinct from non-science can be characterized by its falsifiability: universal statements of theory can be logically contradicted by an intersubjective singular existential statement. Popper was worried that if scientific theories contain conventions that can be freely changed, their criterion of falsifiability cannot be applied. He indicates that a conventionalist can use at least two strategies for preventing falsifiability: (i) introducing ad hoc hypotheses that explain potentially falsifying observations, (ii) changing some ostensive definitions that change the content of the theory.

The first strategy is explicitly used for proposing the dark matter hypothesis. In order to explain the difference in observed rotation velocities of stars in galaxies and galaxies in clusters of galaxies in comparison with theoretical predictions using known theories of gravity and observed masses, it was proposed to assume existence of additional quantities of non-luminous matter, dubbed dark matter. However, the essence and physical properties of dark matter remain mysterious.

Less explicit is the role of introducing hypothetical dark energy for explaining the observed accelerating expansion of the universe. The simplest way to explain it is to complement the Einstein equations of gravitation with a cosmological constant term. It does not change the general interpretation of the theory, but allows cosmological solutions which conform with observations. However, a more detailed interpretation of the cosmological constant is ambiguous [22]. It can be considered as a homogeneous and isotropic perfect fluid, but then its pressure must be negative with an absolute value equal to its energy density. In a sense, the cosmological constant is referred to as a a new typeof perfect fluid.

In both cases, hypothetical entities are introduced for explaining specific observational evidence. Up to now, these entities defy additional observational conformations or refutations. However, they seem to be a solid part of the received view of the universe. As Merritt concludes: "rather than conceive of dark matter and dark energy as postulates invoked in response to falsifying observations—cosmologists interpret those same observations as tantamount to the discovery of dark matter and dark energy".

### 2.2. A Peculiar Feature of Cosmological Science

In cosmological science, there is no possibility to provide experiments or to compare observations of different scenarios. There is only one visible universe and our efforts attempt to find its description. Popper is interested not only in possibilities to falsify a

theory, but even more in finding satisfactory theories. He admits that the aim is in fact not achievable: "Science does not rest on a solid bedrock. The bold structure of its theories rises, as it were, above a swamp. The piles are driven down from above into the swamp, but nor down to any natural or 'given' base, and if we stop driving the piles deeper, it is not because we have reached firm ground. We simply stop when we are satisfied that the piles are firm enough to carry the structure, at least for the time being" [7] (ch.5, sec. 30). In fact, he does not deny that some piles may be conventional, i.e., changeable, although this may be an obstacle to possible falsification of the theory.

In what follows, we argue that there is yet another basic property of the theory of the Friedmann cosmology that can be freely chosen: the geometric framework, a curved torsion-free metric spacetime or a metric spacetime with torsion and flat connections or a flat torsion-free spacetime with nonmetricity. This follows from the fact that the corresponding Lagrangians differ only by boundary terms that vanish in Friedmann cosmology, and local equations of motion can be transformed into each other. This is reminiscent of the geometric conventionality presented by Poincaré [6]: an infinite Euclidean background can be transformed into a finite non-Euclidean background by introducing universal distorting forces. Poincaré concluded that corresponding sets of geometric axioms are just conventions for allowing us to choose a mathematical framework to be applied. The choice is not determined by experiments or observations, although it must be in line with their results.

### 2.3. Empirical Equivalence and Theoretical Equivalence

Although GR, TEGR and STEGR are locally empirically equivalent, this does not mean that they are equivalent in all aspects. There may be other important differences in the theories that do not affect experimental or observational results. Theories may include concepts and theoretical terms that are essentially different; they may attribute different structures to the world, etc. It follows that empirically equivalent but theoretically inequivalent theories may contain conventional parts.

Theoretical equivalence as distinct from empirical equivalence can be described using a formal approach [23], but there are also other ways to consider it. For instance, Coffey [21] proposed that theoretical equivalence of empirically equivalent theories means that they agree on what the physical world is fundamentally like [21]. Note that this is not the view of Poincaré's geometric conventionalism. A short review of different possibilities to introduce theoretical equivalence was recently presented by Weatherall [19]. He admits that empirical equivalence is a necessary condition for two theories to be theoretically equivalent, but need not be a sufficient one.

Dürr [20] considers the case of a theory $T$ together with an empirically equivalent but incompatible alternative account $T'$ of relevant data. Then, $T$ and $T'$ are not simply different representations of the same theory, since they assert contradictory facts about the world. Dürr indicates two ways for this to occur. Firstly, the same mathematical equations can obtain different interpretations, e.g., GR formalism is usually interpreted geometrically, but can also be interpreted field-theoretically, as presented by Feynman [24] and Weinberg [25]. If the inequivalent interpretations are considered to be sufficient for theoretical inequivalence, then we have here two distinct theories of gravitation, although usually physicists consider them to be equivalent with respect to their physical content. Secondly, $T$ and $T'$ can have distinct equations, as in the case of Dirac–von Neumann and Bohmian quantum mechanics, which clearly are two distinct theories, although empirically equivalent.

TEGR and GR describe the underlying spacetime differently although empirically equivalently; in TEGR, mathematical formalism is interpreted as describing a flat spacetime with a non-vanishing torsion, and in GR, the related formalism is interpreted as describing a curved spacetime with a vanishing torsion. In TEGR, the inertial and gravitational forces are described separately, distinct from GR, where they are united via the principle of equivalence. It follows that TEGR and GR are not interpretationally equivalent. At the

same time, the symmetric teleparallel connection endowed with nonmetricity but lacking curvature and torsion arises in STEGR as a Stueckelberg field for diffeomorphisms, meaning it is just a gauge field, i.e., yet another aspect open for interpretation.

Knox [26] argues that if we accept spacetime functionalism, i.e., take into account only those features of the physical world that are functionally relevant in producing empirical evidence, then GR and TEGR can be considered as postulating the same spacetime ontology, since they pick out the same inertial reference frames for gravitational and non-gravitational physics (the limitations of this approach are pointed out by Read and Menon [27]). In Knox's opinion, GR and TEGR are empirically and ontologically equivalent, and they ought to be considered rather as different formulations of the same physics and not two different theories. Her complicated deliberations demonstrate that identity of theories is not an easy problem. Recently, Wolf and Read [28] investigated isolated gravitational systems and systems with boundaries and argued that in this respect, GR and TEGR are equivalent.

If we admit that GR, TEGR and also STEGR can be considered as having equivalent physical content, then we can choose their geometric framework and consider it as conventional. We will return to these issues in Section 4.

## 3. Friedmann Cosmology in Different Formulations of General Relativity

### 3.1. Geometric Preliminaries

To back up and illustrate the discussion, we introduce some maths. In differential geometry, general metric-affine spacetimes are described by two quantities that are in principle independent: the metric $g_{\mu\nu}$, which encodes distances and angles, and the connection $\Gamma^\lambda{}_{\sigma\rho}$, which defines parallel transport and covariant derivatives, e.g.,

$$\nabla_\mu \mathcal{T}^\lambda{}_\nu = \partial_\mu \mathcal{T}^\lambda{}_\nu + \Gamma^\lambda{}_{\alpha\mu} \mathcal{T}^\alpha{}_\nu - \Gamma^\alpha{}_{\nu\mu} \mathcal{T}^\lambda{}_\alpha . \tag{1}$$

The generic affine connection can be decomposed into three parts,

$$\Gamma^\lambda{}_{\mu\rho} = \left\{ {}^\lambda{}_{\mu\nu} \right\} + K^\lambda{}_{\mu\nu} + L^\lambda{}_{\mu\nu} , \tag{2}$$

where the Christoffel symbols of the Levi-Civita connection depend on the metric $g_{\mu\nu}$,

$$\left\{ {}^\lambda{}_{\mu\nu} \right\} \equiv \frac{1}{2} g^{\lambda\beta} \left( \partial_\mu g_{\beta\nu} + \partial_\nu g_{\beta\mu} - \partial_\beta g_{\mu\nu} \right) , \tag{3}$$

while contortion

$$K^\lambda{}_{\mu\nu} \equiv \frac{1}{2} g^{\lambda\beta} \left( T_{\mu\beta\nu} + T_{\nu\beta\mu} - T_{\beta\mu\nu} \right) , \tag{4}$$

and disformation

$$L^\lambda{}_{\mu\nu} \equiv -\frac{1}{2} g^{\lambda\beta} \left( Q_{\mu\beta\nu} + Q_{\nu\beta\mu} + Q_{\beta\mu\nu} \right) \tag{5}$$

encode the independent aspect of the connection. The last two quantities are defined via torsion (antisymmetric)

$$T^\lambda{}_{\mu\nu} \equiv \Gamma^\lambda{}_{\nu\mu} - \Gamma^\lambda{}_{\mu\nu} \tag{6}$$

and nonmetricity (symmetric)

$$Q_{\rho\mu\nu} \equiv \nabla_\rho g_{\mu\nu} = \partial_\rho g_{\mu\nu} - \Gamma^\beta{}_{\mu\rho} g_{\beta\nu} - \Gamma^\beta{}_{\nu\rho} g_{\mu\beta} . \tag{7}$$

Note that torsion and nonmetricity, as well as curvature

$$R^\sigma{}_{\rho\mu\nu} \equiv \partial_\mu \Gamma^\sigma{}_{\nu\rho} - \partial_\nu \Gamma^\sigma{}_{\mu\rho} + \Gamma^\alpha{}_{\nu\rho} \Gamma^\sigma{}_{\mu\alpha} - \Gamma^\alpha{}_{\mu\rho} \Gamma^\sigma{}_{\nu\alpha} \tag{8}$$

and its contractions $R_{\mu\nu} = R^\rho{}_{\mu\rho\nu}$, $R = g^{\mu\nu} R_{\mu\nu}$, are strictly speaking all properties of the connection.

Friedmann cosmology is based on the cosmological principle which expects that at sufficiently large scales, the Universe is homogeneous and isotropic in space, i.e., it is characterised by the Killing vectors of translations $\zeta_{T_i}$ and rotations $\zeta_{R_i}$, given in spherical coordinates as

$$\zeta^{\mu}_{T_x} = \begin{pmatrix} 0 & \chi \sin\theta \cos\phi & \frac{\chi}{r} \cos\theta \cos\phi & -\frac{\chi}{r} \frac{\sin\phi}{\sin\theta} \end{pmatrix}, \tag{9a}$$

$$\zeta^{\mu}_{T_y} = \begin{pmatrix} 0 & \chi \sin\theta \sin\phi & \frac{\chi}{r} \cos\theta \sin\phi & \frac{\chi}{r} \frac{\cos\phi}{\sin\theta} \end{pmatrix}, \tag{9b}$$

$$\zeta^{\mu}_{T_z} = \begin{pmatrix} 0 & \chi \cos\theta & -\frac{\chi}{r} \sin\theta & 0 \end{pmatrix}, \tag{9c}$$

$$\zeta^{\mu}_{R_x} = \begin{pmatrix} 0 & 0 & \sin\phi & \frac{\cos\phi}{\tan\theta} \end{pmatrix}, \tag{9d}$$

$$\zeta^{\mu}_{R_y} = \begin{pmatrix} 0 & 0 & -\cos\phi & \frac{\sin\phi}{\tan\theta} \end{pmatrix}, \tag{9e}$$

$$\zeta^{\mu}_{R_z} = \begin{pmatrix} 0 & 0 & 0 & -1 \end{pmatrix}, \tag{9f}$$

where $\chi = \sqrt{1 - kr^2}$ describes the curvature of the 3D space. The symmetry is obeyed when the Lie derivatives of the metric and affine connection along these vectors vanish [29],

$$\pounds_{\zeta} g_{\mu\nu} = 0, \qquad \pounds_{\zeta} \Gamma^{\lambda}{}_{\mu\nu} = 0. \tag{10}$$

For the sake of simplicity, in this paper, let us focus only upon the spatially flat case, where $k = 0$. It is well known that the metric which satisfies this condition is the Friedmann–Lemaître–Robertson–Walker (FLRW), conveniently written as

$$ds^2 = -dt^2 + a(t)^2 \left( dr^2 + r^2 d\theta^2 + r^2 \sin^2\theta d\phi^2 \right), \tag{11}$$

where $a(t)$ is the scale factor that describes the expansion of space. For a connection with the same symmetries, there are different options to satisfy Equation (10), depending on the extra assumptions made about curvature, torsion and nonmetricity, as discussed below. The matter energy momentum tensor consistent with the cosmological symmetry is given in the same coordinates by

$$\mathcal{T}_{\mu\nu} = \begin{pmatrix} \rho(t) & 0 & 0 & 0 \\ 0 & a^2(t)p(t) & 0 & 0 \\ 0 & 0 & r^2 a^2(t)p(t) & 0 \\ 0 & 0 & 0 & r^2 a^2(t)p(t)\sin^2\theta \end{pmatrix}, \tag{12}$$

where $\rho$ is the energy density and $p$ is the pressure of the matter.

### 3.2. General Relativity

In general relativity, one assumes the connection is torsion-free ($T^{\lambda}{}_{\mu\nu} = 0$) and metric-compatible ($Q_{\rho\mu\nu} = 0$), which leaves only the Levi-Civita part $\{^{\lambda}{}_{\mu\nu}\}$ nonvanishing on the right-hand side of Equation (2). The gravitational field as described by spacetime geometry follows Einstein's field equations,

$$\overset{\text{LC}}{R}_{\mu\nu} - \frac{1}{2} g_{\mu\nu} \overset{\text{LC}}{R} = \kappa^2 \mathcal{T}_{\mu\nu}, \tag{13}$$

while the matter constituents obey the continuity equation

$$\overset{\text{LC}}{\nabla}_{\mu} \mathcal{T}^{\mu}{}_{\nu} = 0, \tag{14}$$

which in the case of a massive point particle, leads to

$$m \left( \frac{du^{\mu}}{d\tau} + \{^{\mu}{}_{\rho\sigma}\} u^{\rho} u^{\sigma} \right) = 0. \tag{15}$$

Here, $m$ is the mass, $u^\mu$ the four-velocity and $\tau$ is the proper time of the particle. The last equation is simultaneously the geodesic equation of the metric (giving the shortest distance), as well as the autoparallel curve of the Levi-Civita connection,

$$u^\nu \overset{\text{LC}}{\nabla}_\nu u^\mu = 0 \,, \tag{16}$$

which says that the particle moves "straight" in the direction of the tangent vector $u^\nu$ of its trajectory. The connection coefficients $\{{}^\mu{}_{\rho\sigma}\}$ in the second term of Equation (15) encode both the inertial effects (a fictional force arising when "straight" motion is described in curvilinear coordinates) and gravitational effects (external force accelerating the particle) together, a fundamental insight of Einstein called the equivalence principle.

In cosmology, it is straightforward to compute the Levi-Civita connection components (3) from metric (11)

$$\{{}^\mu{}_{\rho\sigma}\} = \left[ \begin{bmatrix} 0 & 0 & 0 & 0 \\ 0 & a\dot{a} & 0 & 0 \\ 0 & 0 & a\dot{a}r^2 & 0 \\ 0 & 0 & 0 & a\dot{a}r^2\sin^2\theta \end{bmatrix} \begin{bmatrix} 0 & H & 0 & 0 \\ H & 0 & 0 & 0 \\ 0 & 0 & -r & 0 \\ 0 & 0 & 0 & -r\sin^2\theta \end{bmatrix} \right.$$
$$\left. \begin{bmatrix} 0 & 0 & H & 0 \\ 0 & 0 & \frac{1}{r} & 0 \\ H & \frac{1}{r} & 0 & 0 \\ 0 & 0 & 0 & -\sin\theta\cos\theta \end{bmatrix} \begin{bmatrix} 0 & 0 & 0 & H \\ 0 & 0 & 0 & \frac{1}{r} \\ 0 & 0 & 0 & \cot\theta \\ H & \frac{1}{r} & \cot\theta & 0 \end{bmatrix} \right] \,, \tag{17}$$

where the four matrices in columns are labelled by the first index ${}^\mu$, and the entries of the matrices are specified by the last two indices ${}_{\rho\sigma}$. Here, the dot represents a derivative with respect to time $t$, and the Hubble function $H = \frac{\dot{a}}{a}$ measures the relative expansion rate of space. The connection coefficients (17) obey the cosmological symmetry by construction, and the respective Lie derivatives (10) vanish. From the connection, we can further calculate the curvature tensor (8) and its contractions. Substituting these, as well as the matter energy–momentum (12), into Einstein's Equation (13) yields the Friedmann equations,

$$3H^2 = \kappa^2\rho \,, \qquad 2\dot{H} + 3H^2 = -\kappa^2 p \,, \tag{18}$$

and substitution into the continuity Equation (14) yields

$$\dot{\rho} + 3H(\rho + p) = 0 \,. \tag{19}$$

The solutions of these equations for different types of matter combinations (relativistic/radiation, nonrelativistic/dust, cosmological constant, inflaton field, etc.) describe the evolution of the Universe at large scales. The massive particles moving in the Universe follow Equation (15) with the inertia and background expansion encoded in (17).

### 3.3. Teleparallel Equivalent of General Relativity

After accomplishing the remarkably successful geometrisation of the gravitational field in general relativity, Einstein endeavoured to find a unified geometric theory that would also include electromagnetism. In one of his attempts [30,31], he introduced a spacetime with teleparallelism, where the curvature (8) of the connection vanishes and vectors do not change their direction when parallel-transported along a closed loop. Such connections endowed with torsion (6) were first investigated by Weitzenböck a few years before [32]. Realizing that such a theory cannot accommodate electromagnetism properly, Einstein gave up the idea. However, the concept of teleparallel spacetimes was later invoked by Møller in the search for a description of the energy of gravitational fields [33,34], and then revived by Hayashi and Nakano to construct a possible gauge theory for the spacetime translation group [35], which eventually lead to the development of the teleparallel equivalent of general relativity [8,9] and its extensions [36].

In the teleparallel equivalent of general relativity, one assumes the connection $\overset{\text{TP}}{\Gamma}{}^{\lambda}{}_{\mu\nu}$ is "flat" in the sense of identically zero curvature ($\overset{\text{TP}}{R}{}^{\sigma}{}_{\rho\mu\nu} = 0$) and metric-compatible ($\overset{\text{TP}}{Q}{}_{\rho\mu\nu} = 0$). In the decomposition of the connection (2), this introduces some extra torsional components in the Levi-Civita part. It is important to realize that the Levi-Civita components depend on the metric and when considered among themselves can still be characterised by nontrivial curvature. The role of the extra torsional components in the connection is to "compensate" the Levi-Civita connection in making the overall curvature (8) vanish.

By introducing the torsion scalar

$$\overset{\text{TP}}{T} = \frac{1}{2}\overset{\text{TP}}{T}{}^{\rho}{}_{\mu\nu}\overset{\text{TP}}{S}{}_{\rho}{}^{\mu\nu} \tag{20}$$

where the torsion conjugate (or superpotential) is defined as

$$\overset{\text{TP}}{S}{}_{\rho}{}^{\mu\nu} = \overset{\text{TP}}{K}{}^{\mu\nu}{}_{\rho} - \delta^{\mu}_{\rho}\overset{\text{TP}}{T}{}_{\sigma}{}^{\sigma\nu} + \delta^{\nu}_{\rho}\overset{\text{TP}}{T}{}_{\sigma}{}^{\sigma\mu} \,, \tag{21}$$

the field equations of TEGR can be written as follows [37]

$$\overset{\text{LC}}{\nabla}{}_{\rho}\overset{\text{TP}}{S}{}_{(\mu\nu)}{}^{\rho} - \overset{\text{TP}}{t}{}_{\mu\nu} = \kappa^2 \mathcal{T}_{\mu\nu} \,. \tag{22}$$

An interesting aspect of this form is that the symmetric tensor

$$\overset{\text{TP}}{t}{}_{\mu\nu} = \frac{1}{2}\overset{\text{TP}}{S}{}_{(\mu}{}^{\rho\sigma}\overset{\text{TP}}{T}{}_{\nu)\rho\sigma} - \frac{1}{2}g_{\mu\nu}\overset{\text{TP}}{T} \tag{23}$$

appears in the equations in an analogous position as the energy momentum tensor of the matter. We might be tempted to interpret it as the energy momentum of the gravitational field[1], which acts as a self-source to the dynamics, like the nonlinear self-coupling term in the Yang–Mills equations. Although all geometric tensors that enter here are computed from the teleparallel connection, with a clever use of the geometric identities, it is possible to show that Equation (22) exactly matches Einstein's field Equation (13). In other words, when all the terms in Equation (22) are expanded out in full, only the Levi-Civita part of the connection remains, while the torsional components of the connection cancel each other out. Hence, given a matter energy momentum, both GR and TEGR predict exactly the same evolution for the metric field. There is no equation to give the torsional part of the connection independent dynamics.

In the TEGR constructions, the matter sector is typically assumed to remain unaltered, i.e., maintaining couplings to the metric and Levi-Civita connection only. This guarantees that the continuity Equation (14) holds as before[2]. Hence, the massive particles still follow the geodesics of the metric (15), but using the relation (2) we can rewrite it as

$$m\left(\frac{du^{\mu}}{d\tau} + \overset{\text{TP}}{\Gamma}{}^{\mu}{}_{\rho\sigma}u^{\rho}u^{\sigma}\right) = m\overset{\text{TP}}{K}{}^{\mu}{}_{\rho\sigma}u^{\rho}u^{\sigma} \,. \tag{24}$$

This form suggests an interesting interpretation. Namely, the right-hand side with contortion tensor looks like a force term (akin to the Lorentz force in electrodynamics), while the left-hand side says that in the absence of the force, a massive particle will move "straight" along an autoparallel of the teleparallel connection. Still, both GR and TEGR prescribe identical paths for the particle motion through spacetime. Thus, GR and TEGR are equivalent in the sense that they predict the same physical outcomes, but adding an extra connection allows one to present Equations (22) and (24) in a form where interpretation is more in line with the other well-established and understood theories of physics.

Let us take the Friedmann cosmology example. It can be confirmed that the following connection [29]

$$
\overset{\text{TP}}{\Gamma}{}^{\rho}{}_{\mu\nu} = \left[\begin{bmatrix} 0 & 0 & 0 & 0 \\ 0 & 0 & 0 & 0 \\ 0 & 0 & 0 & 0 \\ 0 & 0 & 0 & 0 \end{bmatrix}\begin{bmatrix} 0 & 0 & 0 & 0 \\ H & 0 & 0 & 0 \\ 0 & 0 & -r & 0 \\ 0 & 0 & 0 & -r\sin^2\theta \end{bmatrix}\right.
$$

$$
\left.\begin{bmatrix} 0 & 0 & 0 & 0 \\ 0 & 0 & \frac{1}{r} & 0 \\ H & \frac{1}{r} & 0 & 0 \\ 0 & 0 & 0 & -\sin\theta\cos\theta \end{bmatrix}\begin{bmatrix} 0 & 0 & 0 & 0 \\ 0 & 0 & 0 & \frac{1}{r} \\ 0 & 0 & 0 & \cot\theta \\ H & \frac{1}{r} & \cot\theta & 0 \end{bmatrix}\right] \tag{25}
$$

is teleparallel, as the curvature (8) and nonmetricity (7) are zero and it obeys the cosmological symmetry, whereby the Lie derivatives with respect to the generators of spatial homogeneity and isotropy vanish, Equations (9) and (10). Substituting this connection, the metric (11) and matter (12) into Equation (22) reproduces the Friedmann Equation (17) in general relativity exactly, with

$$
\overset{\text{TP}}{t}{}_{\mu\nu} = \begin{bmatrix} 3H^2 & 0 & 0 & 0 \\ 0 & -\dot{a}^2 & 0 & 0 \\ 0 & 0 & -r^2\dot{a}^2 & 0 \\ 0 & 0 & 0 & -r^2\sin^2(\theta)\dot{a}^2 \end{bmatrix}. \tag{26}
$$

The last quantity vanishes for a constant scale factor $a$, as an expectedly empty space would not be a source of its own evolution.

For the particle motion, the "force" term on the right-hand side of Equation (24) is set by the contortion (4), and can easily be found by the subtraction of (17) from (25),

$$
\overset{\text{TP}}{K}{}^{\mu}{}_{\rho\sigma} = \left[\begin{bmatrix} 0 & 0 & 0 & 0 \\ 0 & -a\dot{a} & 0 & 0 \\ 0 & 0 & -a\dot{a}r^2 & 0 \\ 0 & 0 & 0 & -a\dot{a}r^2\sin^2\theta \end{bmatrix}\begin{bmatrix} 0 & -H & 0 & 0 \\ 0 & 0 & 0 & 0 \\ 0 & 0 & 0 & 0 \\ 0 & 0 & 0 & 0 \end{bmatrix}\right.
$$

$$
\left.\begin{bmatrix} 0 & 0 & -H & 0 \\ 0 & 0 & 0 & 0 \\ 0 & 0 & 0 & 0 \\ 0 & 0 & 0 & 0 \end{bmatrix}\begin{bmatrix} 0 & 0 & 0 & -H \\ 0 & 0 & 0 & 0 \\ 0 & 0 & 0 & 0 \\ 0 & 0 & 0 & 0 \end{bmatrix}\right]. \tag{27}
$$

It should not be a surprise that when there is no expansion ($\dot{a} = 0$) and the metric reduces to Minkowski, the contortion vanishes and the particle will feel no gravitational "force", although there are still inertial effects present in the connection (25) on the left-hand side of (24) since we use spherical coordinates.

### 3.4. Symmetric Teleparallel Equivalent of General Relativity

From the discussion above, it is not hard to envision another option to present an alternative formulation of general relativity by employing nonmetricity instead of torsion, although the history of this idea dates not that far back [53]. In the symmetric teleparallel equivalent of general relativity [10,11,14], one assumes the connection $\overset{\text{STP}}{\Gamma}{}^{\lambda}{}_{\mu\nu}$ is "flat" ($\overset{\text{STP}}{R}{}^{\sigma}{}_{\rho\mu\nu} = 0$) and torsion-free ($\overset{\text{STP}}{T}{}^{\rho}{}_{\mu\nu} = 0$). Thus, in the decomposition of connection (2), we have extra nonmetricity-related components in the Levi-Civita part. Again, the Levi-Civita components considered among themselves can still be characterised by a nontrivial curvature, and the role of the extra nonmetricity components is to make the overall curvature (8) vanish.

By introducing the nonmetricity scalar

$$
\overset{\text{STP}}{Q} = \frac{1}{2}\overset{\text{STP}}{Q}{}_{\rho}{}^{\mu\nu}\overset{\text{STP}}{P}{}^{\rho}{}_{\mu\nu} \tag{28}
$$

where the nonmetricity conjugate (or superpotential) is defined as[3]

$$
\overset{\text{STP}}{P}{}^{\rho}{}_{\mu\nu} = -\frac{1}{2}\overset{\text{STP}}{Q}{}^{\rho}{}_{\mu\nu} + \overset{\text{STP}}{Q}{}_{(\mu}{}^{\rho}{}_{\nu)} + \frac{1}{2}g_{\mu\nu}\overset{\text{STP}}{Q}{}^{\rho\alpha}{}_{\alpha} - \frac{1}{2}\left(g_{\mu\nu}\overset{\text{STP}}{Q}{}_{\alpha}{}^{\rho\alpha} + \delta^{\rho}{}_{(\mu}\overset{\text{STP}}{Q}{}_{\nu)\alpha}{}^{\alpha}\right),
\tag{29}
$$

the field equations of STEGR can be written in the following form (adopted from [54])

$$
\overset{\text{LC}}{\nabla}_{\rho}\overset{\text{STP}}{P}{}^{\rho}{}_{\mu\nu} - \overset{\text{STP}}{t}{}_{\mu\nu} = \kappa^2 \mathcal{T}_{\mu\nu}
\tag{30}
$$

Here, the symmetric tensor

$$
\begin{aligned}
\overset{\text{STP}}{t}{}_{\mu\nu} = &-\overset{\text{STP}}{L}{}^{\rho}{}_{\alpha\rho}\overset{\text{STP}}{P}{}^{\alpha}{}_{\mu\nu} + \overset{\text{STP}}{L}{}^{\alpha}{}_{\mu\rho}\overset{\text{STP}}{P}{}^{\rho}{}_{\alpha\nu} + \overset{\text{STP}}{L}{}^{\alpha}{}_{\nu\rho}\overset{\text{STP}}{P}{}^{\rho}{}_{\mu\alpha} \\
&-\frac{1}{2}\overset{\text{STP}}{P}{}_{\rho\mu\nu}\overset{\text{STP}}{Q}{}^{\rho\sigma}{}_{\sigma} - \frac{1}{2}\overset{\text{STP}}{P}{}_{\nu\rho\sigma}\overset{\text{STP}}{Q}{}_{\mu}{}^{\rho\sigma} + \overset{\text{STP}}{P}{}_{\rho\sigma\mu}\overset{\text{STP}}{Q}{}^{\rho\sigma}{}_{\nu} + \frac{1}{2}g_{\mu\nu}\overset{\text{STP}}{Q}
\end{aligned}
\tag{31}
$$

again appears in the equations in an analogous position to the energy momentum tensor of the matter and we might be tempted to interpret it as a kind of energy momentum of the gravitational field. Although all geometric tensors in the field equations are computed from the teleparallel connection, with a clever use of the geometric identities, it is possible to show that Equation (31) matches Einstein's field Equation (13) exactly. Indeed, when all the terms in Equation (22) are expanded out in full, only the Levi-Civita part of the connection remains, while the nonmetricity components of the connection cancel each other out. Hence, given a matter energy momentum, both GR and STEGR predict exactly the same evolution for the metric field. There is no equation to give the nonmetricity part of the connection independent dynamics.

Leaving the matter sector in STEGR unaltered from GR, i.e., maintaining couplings to the metric and Levi-Civita connection only[4], guarantees that the continuity Equation (14) holds as before. Hence, the massive particles still follow the geodesics of metric (15), but using the relation (2), we can rewrite it as

$$
m\left(\frac{du^{\mu}}{d\tau} + \overset{\text{STP}}{\Gamma}{}^{\mu}{}_{\rho\sigma}u^{\rho}u^{\sigma}\right) = m\overset{\text{STP}}{L}{}^{\mu}{}_{\rho\sigma}u^{\rho}u^{\sigma}.
\tag{32}
$$

Analogously to the torsional case, the right-hand side with the disformation tensor looks like a force term, while the left-hand side says that in the absence of the force, a massive particle will move "straight" along an autoparallel of the symmetric teleparallel connection. Yet, both GR and STEGR prescribe identical paths for the particle motion through spacetime. Therefore, GR and STEGR are equivalent in the sense that they predict the same physical outcomes, but adding an extra connection allows one to present Equations (31) and (32) in a form where interpretation is more in line with the other well-established and understood theories of physics.

The options for the symmetric teleparallel connection that obey cosmological symmetry (9) and (10) were determined in refs. [56,57]. It turns out that for spatially flat connections there are three sets of solutions. Perhaps the simplest of those is set 1 (in the notation of refs. [58,59])

$$
\overset{\text{STP}}{\Gamma}{}^{\rho}{}_{\mu\nu} = \left[\left[\begin{matrix}\gamma(t) & 0 & 0 & 0 \\ 0 & 0 & 0 & 0 \\ 0 & 0 & 0 & 0 \\ 0 & 0 & 0 & 0\end{matrix}\right]\left[\begin{matrix}0 & 0 & 0 & 0 \\ 0 & 0 & 0 & 0 \\ 0 & 0 & -r & 0 \\ 0 & 0 & 0 & -r\sin^2\theta\end{matrix}\right] \\ \left[\begin{matrix}0 & 0 & 0 & 0 \\ 0 & 0 & \frac{1}{r} & 0 \\ 0 & \frac{1}{r} & 0 & 0 \\ 0 & 0 & 0 & -\sin\theta\cos\theta\end{matrix}\right]\left[\begin{matrix}0 & 0 & 0 & 0 \\ 0 & 0 & 0 & \frac{1}{r} \\ 0 & 0 & 0 & \cot\theta \\ 0 & \frac{1}{r} & \cot\theta & 0\end{matrix}\right]\right]
\tag{33}
$$

where $\gamma(t)$ is a free function. Substituting this connection, metric (11) and matter (12) into Equation (31) reproduces the Friedmann Equation (17) in general relativity exactly, with

$$
\overset{\text{STP}}{t}{}_{\mu\nu} = \begin{bmatrix} 3\gamma H & 0 & 0 & 0 \\ 0 & -a^2(2H^2 + \gamma H) & 0 & 0 \\ 0 & 0 & -a^2 r^2(2H^2 + \gamma H) & 0 \\ 0 & 0 & 0 & -a^2 r^2 \sin^2(\theta) a^2(2H^2 + \gamma H) \end{bmatrix}. \tag{34}
$$

Again, the last quantity vanishes for s constant scale factor $a$, which is consistent with the expectation that empty space does not act as a source of itself. Although the free function $\gamma(t)$ enters (34), it cancels out in the full field in Equation (31). This is not a surprise, as this function is not present in GR, and GR equations do not contain anything to determine it.

For the particle motion, the "force" term on the right-hand side of Equation (32) is set by the disformation and can be found by computing (5) from (33), yielding

$$
\overset{\text{STP}}{L}{}^{\mu}{}_{\rho\sigma} = \left[ \begin{bmatrix} \gamma & 0 & 0 & 0 \\ 0 & -Ha^2 & 0 & 0 \\ 0 & 0 & -r^2 Ha^2 & 0 \\ 0 & 0 & 0 & -r^2 Ha^2 \sin^2(\theta) \end{bmatrix} \begin{bmatrix} 0 & -H & 0 & 0 \\ -H & 0 & 0 & 0 \\ 0 & 0 & 0 & 0 \\ 0 & 0 & 0 & 0 \end{bmatrix} \right.
$$
$$
\left. \begin{bmatrix} 0 & 0 & -H & 0 \\ 0 & 0 & 0 & 0 \\ -H & 0 & 0 & 0 \\ 0 & 0 & 0 & 0 \end{bmatrix} \begin{bmatrix} 0 & 0 & 0 & -H \\ 0 & 0 & 0 & 0 \\ 0 & 0 & 0 & 0 \\ -H & 0 & 0 & 0 \end{bmatrix} \right]. \tag{35}
$$

Interestingly, when there is no expansion ($\dot{a} = 0$) and the metric reduces to Minkowski, the disformation will still depend on the arbitrary function $\gamma$. This does not affect the actual particle trajectory, though, as in the equations of particle motion (32), the $\gamma$ terms drop out. We may take $\gamma$ to vanish identically without any constraints from the equations. This just illustrates how adding the non-Riemannian part to make the overall connection vanish contains an aspect of arbitrariness, as the split between inertia and force is up to our liking at this stage.

The two other symmetric teleparallel connections that obey the cosmological symmetries are rather similar [56,57]. They both introduce an arbitrary function, which by assumption cannot be zero. The cosmological field equations and particle motion equations are quite analogous to the previous case, and do not introduce qualitatively new features for our purposes. It can just be remarked that the free function $\gamma$ of set 1 does not appear in the cosmological equations of extended symmetric teleparallel theories either [56,57], but the functions present in the alternative sets 2 and 3 become dynamical and can easily trigger a finite-time singularity in the extended theories [59].

*3.5. General Teleparallel Equivalent of General Relativity (GTEGR)*

Besides only activating torsion, as in teleparallel gravity, or only activating nonmetricity, as in symmetric teleparallel gravity, one may also entertain the option of having both a torsion and nonmetricity different from zero, while the curvature is still restricted to vanish. In such a setting, it is possible to formulate the general teleparallel equivalent of general relativity which gives the same equations and predictions as GR [18]. For this theory, the connections with cosmological symmetries were determined in ref. [60], and considerations in relation to the notion of energy were given in ref. [61]. In view of the present paper, GTEGR does not add qualitatively new features, and we will not go deeper into the details here.

## 4. Discussion

In this paper, we argue that there are at least three conventional elements in the standard $\Lambda$CDM cosmology; in addition to (1) dark matter and (2) dark energy, there is also (3) the type of geometry. The conventionality of the first two entities is discussed by

Merritt [4]. They are introduced with the aim of evading consequences of observations that can falsify the standard ΛCDM cosmology. However, in principle, it is possible that dark matter particles can be detectable by some non-gravitational means, or the effects otherwise attributed to dark matter can be explained by a suitable (ultra-low acceleration) modification to the gravitational force law testable in some other experiments as well. Similarly, it is conceivable that the source of dark energy can be independently identified as some new classical field or a quantum field theory or quantum gravity effect. Thus, in principle, it is possible to break the ad hoc nature of dark matter and dark energy. The character of the third element seems to be different however.

The third convention emerges if we take into account that local properties of the standard cosmology can be given in different geometrical frameworks that are observationally equivalent. We can write the same equations in GR with a Levi-Civita connection that has curvature in TEGR with a teleparallel connection that has an identically zero curvature but nontrival torsion; in STEGR with a symmetric teleparallel connection that is endowed with nonzero nonmetricity but vanishing curvature; or in GTEGR with a general teleparallel connection where the curvature is zero but both torsion and nonmetricity can be present. Despite invoking these different geometric structures, Einstein's field equations and particle motion equations reduce to the same immediate mathematical content in all formulations and predict the same physical outcomes for given initial conditions.

The situation is reminiscent of the geometric conventionality presented by Poincaré [6], who envisioned how an infinite Euclidean background can be transformed into a finite non-Euclidean background by introducing universal distorting forces. Poincaré concluded that the corresponding sets of geometric axioms are neither synthetic a priori nor experimental facts; they are conventions that allow us to choose the mathematical framework to be applied. The choice is not determined by experiments or observations, although it must be in line with their results and, last but not least, must avoid contradictions.

In the modern teleparallel version of geometric conventionalism, as we saw in Section 3, the split between the "source" and "kinetic" terms of the Einstein's field equations, or the "inertia" and "force" terms in the particle motion equations, is a convention, up to the choice of the formulation or different choices of connection classes within a given formulation. It may even be up to the choice of an arbitrary function in a particular class of connections within a given formulation. To be more precise, the metric structure in geometry can be related to a physical observable as giving a distance between spacetime points. It also defines the light cones that determine which points can be causally connected. Thus, the different formulations of GR do not diverge about the metric, otherwise the empirical equivalence would be broken. On the other hand, the different formulations prescribe different connections in setting up the underlying geometry. The basic role of the connection is to define which path is "straight", whereby any physicist immediately recalls Newton's first law. However, as soon as one departs from the absolute space of Newton and intermingles gravity with geometry, the "straightness" of particle motion becomes ambiguous. Even if for one connection some path (a collection of points) is "straight", i.e., autoparallel as defined by that connection, for another connection, the same path (the same collection of points) is not "straight" any more, while the deviation can be attributed to a corresponding "force" acting on the particle. Empirically, what is available for observation is the path, not its "straightness". The choice of the connection is a convention, in spite of how contrary to the usual GR intuition this statement is. If we can choose which type of geometry we use, then none of them can be considered as describing the "real" spacetime, at least from the point of view of local equations of motion.

At this point, one may object that the Levi-Civita connection is the unique connection obtainable from the metric and it is the minimal but sufficient choice to describe all phenomena within the purview of GR and its empirical equivalents. Hence, to keep the list of agents in the game as short as possible, by the principle of Occam's razor, we can drop the extra non-Riemannian parts of connection as they are superfluous, do not contribute any observable effects and are not even determined by the equations. While this view has

its merits, a counterargument can compare the choice of the connection to the choice of a gauge in electrodynamics. Although different gauge choices of the vector potential imply the same electric and magnetic field strengths and the same motion for a charged particle, in some gauges, the practical computations can become much more economical to perform. Similarly, it may happen that a good choice of the connection can considerably simplify the gravitational calculations. For example, in symmetric teleparallelism, there always exists a coordinate system where the connection vanishes identically in the whole spacetime, and thus the covariant derivatives reduce to plain partial derivatives, called a 'coincident gauge' [10].[5] In practical terms, if picking a good extra connection would help in running the numerical simulations in gravity or assist in a consistent quantisation of gravity, the blade of Occam's razor could be turned the other way.

Given that GR, TEGR, STEGR and GTEGR are empirically equivalent, it remains to ask whether they are also theoretically equivalent? The answer, of course, depends on the philosophical definition of the notion of theoretical equivalence. Instead of delving deeply into that discussion, let us instead mention a few aspects from a more physical point of view. Since the theories are still under active investigation and development, the following remarks only reflect the present state of understanding (of the authors).

First, the fundamental theories of physics obey the action principle, whereby a single mathematical expression encodes all information about the theory, as the equations, conserved quantities and other features can be derived from it. The actions of GR and those of TEGR, STEGR and GTEGR differ by their respective boundary terms [10,11,14]. This means that their field equations are fully equivalent in the usual spacetimes without boundaries, as in the case of Friedmann cosmology. That explains why Equations (13), (22) and (30) match. In more complicated situations of spacetimes with boundaries like the braneworld models, the presence of a boundary might require extra source terms in the action. However, the question of how the correspondence between the different formulations actually works out in that case has not received much attention yet. In addition, there is a broader issue of whether the equations of motion are all that are relevant in physics. While the boundary term in the action does not affect the field equations in the bulk, it can still play a role in something. For example, the correct account of the black hole entropy and thus the establishment of generalised thermodynamics wholly depends on the boundary term in the GR action [28]. The investigation into black hole thermodynamics in the teleparallel context has barely begun [45,46,63,64], and we do not know whether a consistent account can be given in all the formulations.

The second issue is the well-established problem of gravitational energy, where Noether's theorem applied to GR does not yield a local quantity that would be both covariant and conserved [65]. At best, one can entertain global integral quantities for asymptotically flat spacetimes. Since the early days [33], the hope of finding a consistent definition of energy for the gravitational field has been one motivating factor in the investigations of teleparallel theories, and there are different proposals and arguments in the recent literature [38–51]. At the present moment, there is not yet a consensus on whether a universal definition of gravitational energy–momentum can be given, how it is given and whether it can be given in all or only in a specific formulation of the otherwise empirically equivalent family of formulations. Although different assignments of energy to spacetime configurations may not alter any of the observable predictions, it could be that a certain formulation of the theory is preferable in terms of elegance and consistency with the rest of physical theories. Or it could be that in the end, all formulations turn out to be equivalent also in this respect.

Third, the equivalence discussed so far concerns just the local properties of the theory, represented by the field equations and motion of test particles. The global properties of the corresponding spacetimes, including the effects of topology, have not yet been investigated much. It is a well-known fact in topology that not all spaces admit global parallelization. Even in the case of Friedmann cosmology, the metric (11) is compatible with different topologies by clever global indentifications, and such scenarios are not completely ruled

out by current observations [66]. Thus, it remains an open problem how the teleparallel constructions would fare in nontrivial topologies, and whether the equivalence would still stand.

## 5. Conclusions

We think that it is not surprising that our cosmological standard model contains so many parts that are fixed by conventions. Much more surprising is the fact that not all of our cosmological knowledge is a conventional narrative only.

**Author Contributions:** Investigation, validation, formal analysis, supervision, writing—review, editing, funding acquisition, project administration: L.J. and P.K.; conceptualization, methodology, writing—original draft preparation: P.K. All authors have read and agreed to the published version of the manuscript.

**Funding:** The work was supported by the Estonian Research Council grant PRG356 "Gauge Gravity". The authors also acknowledge support from the European Regional Development Fund through the Center of Excellence TK133 "The Dark Side of the Universe".

**Data Availability Statement:** Data are contained within the article.

**Conflicts of Interest:** The authors declare no conflict of interest.

## Abbreviations

The following abbreviations are used in this manuscript:

| | |
|---|---|
| FLRW | Friedmann–Lemaître–Robertson–Walker |
| GR | general relativity |
| GTEGR | general teleparallel equivalent of general relativity |
| STEGR | symmetric teleparallel equivalent of general relativity |
| TEGR | teleparallel equivalent of general relativity |

## Notes

1    The issue of gravitational energy-momentum in teleparallel gravity has seen quite some developments and arguments [38–51], but we do not intend to make a strong claim here.

2    Modification of the matter sector with added couplings to the non-Riemannian part of the connection typically introduces new terms in the continuity equation and particle motion equation [52], breaking the equivalence with GR.

3    Often in the literature the factor $\frac{1}{2}$ in (28) is moved into the definition (29).

4    For simple scalar, spinor and vector fields, we may actually replace the Levi-Civita connection with the symmetric teleparallel connection [55].

5    In these coordinates, the metric typically becomes more complicated though, which can considerably curb the benefits in calculational economy [62].

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
