# Peer review of "Conventionalism, Cosmology and Teleparallel Gravity"

_universe, doi:10.3390/universe10010001_

Round 1

Reviewer 1 Report

Comments and Suggestions for Authors

The authors study the equivalence/convention in the interpretation of cosmology using three geometrically different but physically equivalent descriptions of gravity, i.e. General Relativity based on the curvature, Teleparallel gravity based on the torsion and Symmetric teleparallel gravity based on the non-metricity of the connection of spacetime.

The authors focus on the cosmology predicted by these theories and they claim that the choice of the picture we decide to use is a convention. The paper is very interesting and it refers to very nice papers in the literature. 

I would suggest the publication of the paper in its current form, but I found some confusing statements that would better be clarified for the reader.

1) In the 2nd paragraph of the introduction, it is said that DE was introduced to explain the accelerating expansion of the Universe (correct) and that DM was introduced to explain the rotation curves of galaxies (partially correct). There are more hints for DM like gravitational lensing and CMB. Either rephrase or add these as well.

2) The last sentence of the 4th paragraph of Section 2.3 says "the same holds for STEGR". This is not entirely correct. The use of an arbitrary spin connection in TEGR can indeed separate inertial from gravitational effects. However, in STEGR the connection acts as a Stuckelberge field for diffeomorphisms, meaning that it is just a gauge field and thus it cannot play any physical role. So the fact that inertial effects are separated from gravitational ones, is not correct in STEGR.

3) The last paragraph of Sec. 2.3 (and even the second to last) discussed that the three theories describe the same physics but are not different theories. This is indeed an interesting way to think about this trinity. However, purely from a theoretical point of view, if we use the Ocam's razor, both TEGR and STEGR are superior than GR, since they do not need the principle of equivalence to be formulated.

4) Last but not least, all this analysis and the nice discussion not only is based on cosmology but also on FLRW cosmology. This means that the "conventionalism" is strictly constrained at very large scales and lower scales physics could distinguish between the three interpretations. Apart from that, once we consider even the slightest deviation from homogeneity and/or isotropy this whole discussion is moot.

All in all, as I mentioned above, the paper is worth publishing, but I think the above points would make its points even stronger. 

Author Response

We thank the referee for relevant comments and suggestions. In the updated manuscript we have

1) extended the sentence on the motivation for dark matter (line 21),

2) altered the sentence on the interpretation of STEGR (lines 143-143),

3) added a sentence on line 159 that refers to Sec. 4 which harbors a more extensive discussion on Occam's razor (see lines 350-366),

4) added a phrase to line 56-57, drawing extra attention that while the primary focus of the paper is upon FLRW cosmology, the overall discussion is wider with links to black hole thermodynamics, gravitational energy, and topology in general (see lines 372-412).

Reviewer 2 Report

Comments and Suggestions for Authors

The author in this paper believes that GR, TEGR, STEGR, and GTEGR are empirically equivalent, and whether they are theoretically equivalent depends on the philosophical definition of the notion of theoretical equivalence. It is indicated that these different formulations of general relativity are locally equivalent, in the sense they predict the same physical outcomes. The corresponding cosmological models are equivalent and their choice is conventional. The problems of theoretical and interpretational equivalence in physics are discussed. Some fundamental topic, the gauge choices, the action principle and the problem of gravitational energy are also discussed, some suggestive ideas are proposed. In my opinion, it is a good critical review.

Author Response

We thank the referee for reviewing the manuscript.

Reviewer 3 Report

Comments and Suggestions for Authors

The authors discuss the issue of conventionalism in the context of three modified gravities, GR, the teleparallel version of GR and the symmetric teleparallel equivalent of GR. I found the manuscript with a well introduction, a proper structure and very interesting discussion. The mentioned theories have nicely been reviewed. They have also used up-to-date references. Hence, I suggest the paper to be published in Universe in the present form.

Author Response

(The authors gave the same response as above.)

Reviewer 4 Report

Comments and Suggestions for Authors

Authors consider the Friedmann cosmological model from three points of view. First, they describe this model in the "usual", metric general Eisteinien relativity and then in the theory with teleparallelism with torsion and without torsion. The main conclusion of the paper is that all those formalisms lead to equivalent results and the choice of one of them is a convention only. I agree with that conclusion but I suppose that one should also take into account the criterion of simplicity and elegance. From this point of view the original Einstein metric formulation seems the best one. However, it is worth analyzing others  mathematical models from the practical viewpoint what is discussed in Sec. 4 of the reviewed paper. 

Concluding: although the work does not contain very revealing ideas it can be valuable for both the physicists and  philosophers of science and I recommend it for publication as it stands now.

Author Response

(The authors gave the same response as above.)
